# Network-based visualisation reveals new insights into transposable element diversity

Lisa Schneider[1,2,3], Yi-Ke Guo[3,†], David Birch[3] & Peter Sarkies[1,2,*] iD

## Abstract

Transposable elements (TEs) are widespread across eukaryotic genomes, yet their content varies widely between different species. Factors shaping the diversity of TEs are poorly understood. Understanding the evolution of TEs is difficult because their sequences diversify rapidly and TEs are often transferred through non-conventional means such as horizontal gene transfer. We developed a method to track TE evolution using network analysis to visualise TE sequence and TE content across different genomes. We illustrate our method by first using a monopartite network to study the sequence evolution of Tc1/mariner elements across focal species. We identify a connection between two subfamilies associated with convergent acquisition of a domain from a protein-coding gene. Second, we use a bipartite network to study how TE content across species is shaped by epigenetic silencing mechanisms. We show that the presence of Piwi-interacting RNAs is associated with differences in network topology after controlling for phylogenetic effects. Together, our method demonstrates how a network-based approach can identify hitherto unknown properties of TE evolution across species.

**Keywords** epigenetics; evolution; networks; piRNAs; transposable elements
**Subject Categories** Chromatin, Transcription & Genomics; Computational Biology; Evolution & Ecology
**Mol Syst Biol. (2021) 17: e9600**

## Introduction

Transposable elements (TEs) are ubiquitous components of eukaryotic genomes. Several subtypes of TEs exist, broadly characterised into type I TEs, which replicate via an RNA intermediate, and type II TEs, which replicate at the DNA level (Wicker *et al*, 2007). Although both types are present in almost all eukaryotic genomes so far characterised, the specific TE content of genomes varies hugely both in terms of the diversity of TE subtypes present and in terms of the percentage TE content across the genome. The factors driving the diversity of TE content across genomes are largely unclear (Chuong *et al*, 2017).

One important factor potentially shaping TE diversity is the control mechanisms employed to limit their spread. Due to their ability to copy themselves independently of the replication of the host, TEs have earned a reputation as selfish elements: parasites of the genome with potentially disadvantageous connotations for organismal fitness (Bourque *et al*, 2018). Unchecked TE proliferation poses several threats to genomic integrity including gene disruption, ectopic recombination between homologous TEs at different chromosomal loci and competing with endogenous genes for cellular resources (Bourque *et al*, 2018). Thus, organisms are under continued pressure to control transposable elements and have evolved sophisticated strategies to recognise TEs and target them for silencing. However, these strategies evolve rapidly and differences in the silencing mechanisms employed might be important factors influencing TE content in genomes.

Epigenetic silencing mechanisms are frequently involved in silencing TEs. Epigenetic mechanisms provide particularly robust TE silencing since they have the ability to propagate through cell division independent of the initiating signal (Bird, 2002). They are often widely conserved across eukaryotes (Law & Jacobsen, 2010). One ancient silencing mechanism is cytosine DNA methylation, which is targeted to TEs in mammals, plants, some nematodes (Rainey *et al*, 2016; Rošić *et al*, 2018) and some arthropods(de Mendoza *et al*, 2019). The mammalian DNA methyltransferases are the enzymes DNMT1, DNMT3A and DNMT3B (Law & Jacobsen, 2010). However, in many species DNA methylation is not found at TEs and instead is targeted to the exons of expressed genes (Feng *et al*, 2010; Zemach *et al*, 2010; Bewick *et al*, 2017, 2019). Moreover, DNA methylation has been lost altogether many times independently in eukaryotic lineages (Ponger & Li, 2005; Feng *et al*, 2010; Zemach *et al*, 2010; Jurkowski & Jeltsch, 2011; Rošić *et al*, 2018).

In metazoans, another key TE silencing mechanism is the Piwi-interacting RNA pathway, whereby small (20–33 nucleotide) RNAs (piRNAs) associate with a protein of the Piwi subfamily of Argonaute proteins to recognise TEs and target them for transcriptional and post-transcriptional silencing. Although widely conserved, piRNAs have been lost a number of times independently in nematodes (Sarkies *et al*, 2015) and were also lost in parasitic flatworms (Skinner *et al*, 2014) and dust mites (Mondal *et al*, 2018).

1  MRC London Institute of Medical Sciences, London, UK
2  Institute of Clinical Sciences, Imperial College, London, UK
3  Data Sciences Institute, Imperial College, London, UK
  *Corresponding author. Tel: +44 7570790430; E-mail: psarkies@imperial.ac.uk
  †Present address: Hong Kong Baptist University, Kowloon Tsai, Hong Kong

Despite the threats posed by TE mobilisation, it is clear that TE insertions have been important in shaping adaptive evolution, for example promoting shuffling of exons and generating new regulatory regions (Chuong *et al*, 2017). The balance between potentially beneficial and deleterious effects of TEs may be another factor influencing their diversity.

A third important factor that shapes TE content across genomes is genomic drift where TE content in a genome reflects population history rather than particular control mechanisms or adaptive functions. Drift is likely to be the major factor determining the persistence of selectively neutral changes in TE copy number. In small population sizes, even deleterious changes in TE content can still occur through drift because the efficiency of natural selection is much weaker (Lynch & Conery, 2003). Genomic drift is a "null hypothesis" to explain why TE content might differ between species (Lynch & Conery, 2003; Szitenberg *et al*, 2016).

So far, our understanding of the balance between TE silencing, adaptive benefit of TEs and genomic drift in shaping differences in TE content is largely theoretical. In order to test competing explanations against the null hypothesis of genetic drift, information about TE content across phylogenies is required. This offers the opportunity to study how repeated events such as loss of piRNAs or loss of DNA methylation affect TE content. However, performing these analyses is challenging because TE sequences evolve very rapidly across genomes, and each genome contains a highly complex series of often unique TE families (Wicker *et al*, 2007). Where attempts to analyse TE content across phylogenies have been performed, they have focussed on broad measures of TEs such as percentage coverage of major classes and have concluded that drift is the major force shaping such values (Szitenberg *et al*, 2016).

Here, we develop a new approach to compare TE diversity across species using network analysis. Inspired by a network approach to analyse virus evolution (Iranzo *et al*, 2016), we create networks to visualise TE evolution across different genomes. Our approach leverages the information within genomes to improve visualisation and make it easier to discern differences in TE type or content than traditional sequence alignment methods or broad investigations of genomic TE content. We used monopartite networks to analyse the evolution of the Tc1/mariner superfamily of TEs, which led to discovery of a hitherto unappreciated connection between two Tc1/mariner subfamilies, due to the parallel acquisition of a domain derived from a protein-coding gene. We also built bipartite networks comparing the coverage of different categories of TEs across a wide phylogeny to investigate how TE content evolves across metazoans. We show that the presence of piRNAs in particular has significant effects on the properties of this bipartite network, which prompts hypotheses about the roles that TE silencing mechanisms play in shaping TE content across evolutionary time. Our work provides a new tool for visualising genome evolution across millions of years.

## Results

### Constructing a sequence similarity network of the Tc1/mariner superfamily

Network visualisation is a powerful tool to understand interactions (Charitou *et al*, 2016; Iranzo *et al*, 2016). We decided to apply a similar approach to visualise transposable element content across genomes. We first applied our method to alignments between transposable elements drawn from the Tc1/mariner superfamily. We chose this TE family due to its wide phylogenetic distribution (Dupeyron *et al*, 2020), and because Tc1/mariner elements are well characterised as active TEs in *C. elegans* (Bessereau, 2006). We constructed a network in which each individual TE is a separate node and the weight of the connection between each node is determined by the sequence similarity. We identified mariner TEs using RepeatMasker from 5 genomes across nematodes with the arthropod *Drosophila melanogaster* as an out-group (Fig 1A). Each node represents a single transposable element sequence, and the strength of the connection between any two nodes represents the BLAST bit score when the two are aligned. Our approach differs from the approach used by Iranzo et al. to investigate viral genomes (Iranzo *et al*, 2016) because in our network, connections are governed by the similarity between complete TE sequences at the nucleotide level rather than through comparison of the encoded proteins only. It is important to note that classical phylogenetic approaches such as construction of phylogenetic trees would not be informative as the sequence similarity across the Tc1/mariner family is too weak. As a result, phylogenetic trees even encompassing small subfamilies give poorly resolved trees with considerable uncertainty. Additionally, our approach offers the opportunity to identify sequence similarity that does not coincide with homology, such as convergent evolution of particular regions within the TE.

We visualised the Tc1/mariner network using the OpenOrd layout algorithm that attempts to bring nodes together into clusters based on the strength of the connection between them. The network formed several closely connected clusters (Fig 1B). To investigate the composition of these clusters, we used two distinct algorithms implemented in Gephi: the connected component statistic implemented by the depth-first search algorithm and the modularity calculation using the Louvain Method for community detection (Tarjan, 1972; Blondel *et al*, 2008; Bastian *et al*, 2009). Both methods identified several distinct clusters of Tc1/mariner elements (Fig 1B; Appendix Fig S1A and B). The rand index, which quantifies the fraction of TEs in the same group in both methods, was 0.6, suggesting good agreement. Most of the clusters correspond to TEs from one subfamily within either one species or two closely related genomes (Fig 1C and D). To test whether the clustered structure of the network was specific to the choice of species, we constructed an additional Tc1/mariner network using 5 arthropod species and *C. elegans* as an outlier (Materials and Methods). This network also showed clear clusters qualitatively similar to the nematode network (Appendix Fig S2A–C).

Tc1 elements are class II DNA transposable elements which move by a cut-and-paste mechanism. Autonomous active TEs contain a DDE3 domain required for mobility. Many Tc1 fragments within genomes no longer retain the potential to spread due to acquisition of inactivating mutations. Interestingly, across the network Tc1/mariner elements with a DDE3 domain show higher connectivity than those without DDE3 domains ($P < 0.001$) (Fig 1E and F). The presence of the DDE3 domain is required for activity; thus, the higher connectivity may represent more recent activity within genomes resulting in a larger number of closely related copies. We did not observe a significant difference in connectivity for DDE3 containing TEs within the arthropod Tc1/mariner network ($P = 0.21$); thus, this result may be specific for nematodes (Appendix Fig S2E and F).

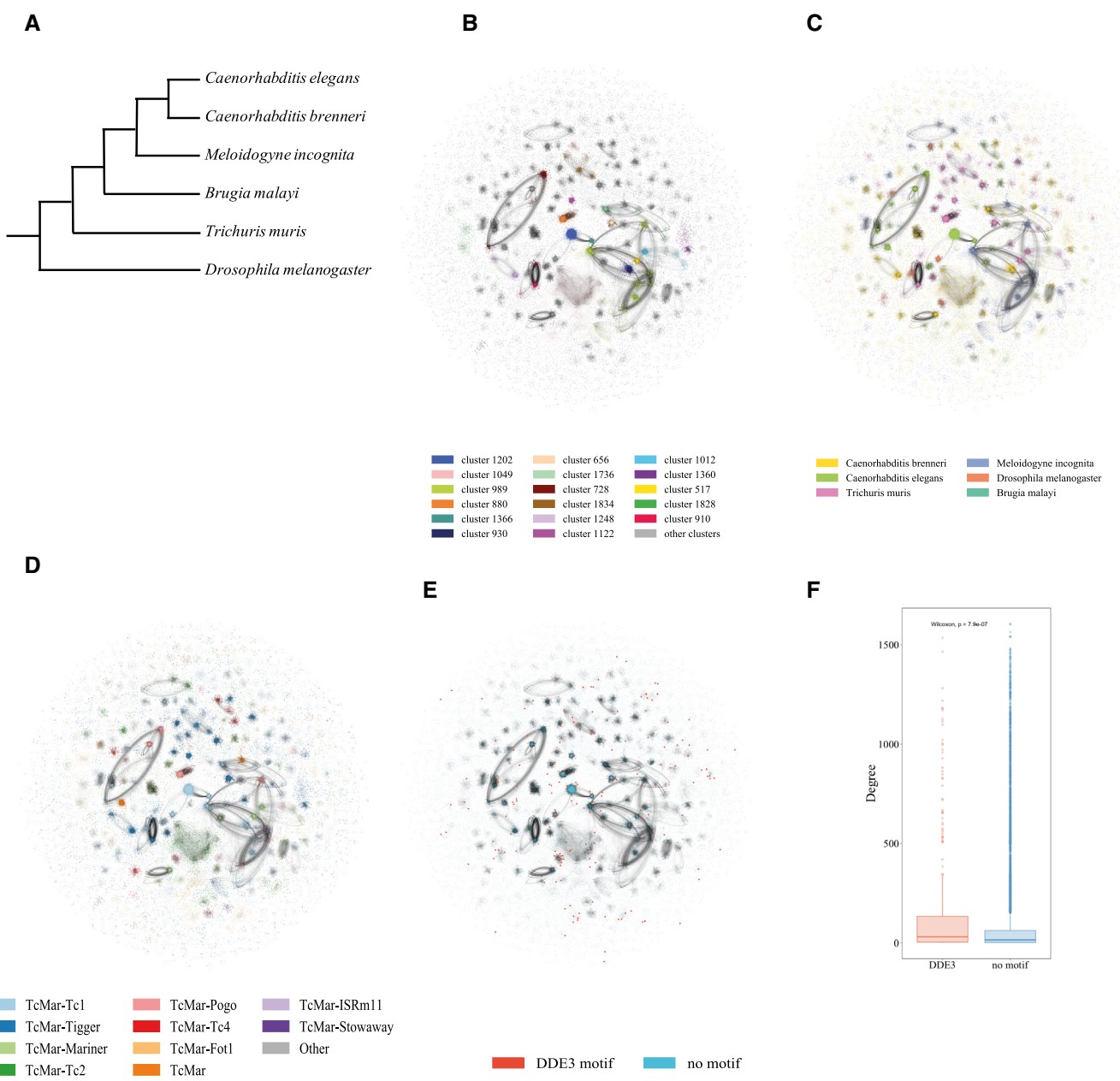

**Figure 1. TcMar/mariner sequence similarity network.**

A   A phylogenetic tree showing the evolutionary relationship between the six species in the Sequence Similarity Network (SSN).

B   Visualisation of the network using the openOrd layout algorithm. The SSN consists of the five nematodes *B. malayi*, *C. elegans*, *M. incognita*, *C. brenneri* and *T. muris* and the insect *D. melanogaster*. Each node in the network represents a TE copy from the respective genome, and the edge weights indicate sequence similarity. The nodes belonging to the 14 largest clusters are coloured by cluster identity, and all other nodes are coloured in grey.

C   The SSN with all nodes coloured according to the sequences' respective origin genome of origin.

D   The SSN with nodes coloured according to the subfamily of the TE.

E   The sequences with a DDE3 transposase domain are highlighted (orange).

F   Boxplot showing the difference in connectivity between 315 sequences containing a DDE3 motif and 21,940 sequences without a motif. DDE3 containing sequences have a significantly higher connectivity (*P* < 0.001, Wilcoxon). The box outlines the interquartile range (IQR) with the median shown as a line, whiskers indicate the fences (lower fence: lowest value at most Q1−1.5*IQR, upper fence: largest value no further than Q3+1.5*IQR) and data beyond the whiskers are drawn as individual points (outliers).

## Network properties illuminate cryptic sequence features of TEs

Within our sequence similarity network, the majority of clusters were formed of one known subfamily of Tc1/mariner TEs (Fig 2A–C). However, a small number of clusters contained more than one subfamily. As an example, we selected one of these clusters, mc476, which was composed of two subfamilies annotated by our pipeline as Fot1 and Tc1 (Fig 2D). Fot1 TEs are most common in fungi, although some metazoan examples have been found (Dupeyron *et al*, 2020). The origin of this element in nematodes is thus unknown. We did a phylogenetic alignment of all sequences in the mixed cluster and four out-group sequences classified as the same Fot1 element type. The tree does not support a common origin for the elements within the cluster, suggesting that homology was not the reason for the co-occurrence of these TEs within the network (Appendix Fig S3). We therefore speculated that there might be a cryptic region of sequence similarity within the two elements responsible for the clustering. To test this hypothesis, we separately blasted all the elements in the cluster against the *C. elegans* proteome. Intriguingly, 53 out 63 sequences had a best blast hit to the nematode Rabconnectin-3 protein (*C. elegans rbc-1*) and we identified RAVE domains in many of them(Fig 2E–G). We observed that *rbc-1* hits were significantly less common in sequences outside of the mixed cluster (Fig 2H). *rbc-1* is highly conserved and in yeast has been shown to assemble the Vacuolar (V-type) ATPase complex required for endosome acidification (Smardon *et al*, 2002), with similar roles suggested in *Drosophila* (Yan *et al*, 2009) and mammals, where the homologue is named DMXL2 (Sethi *et al*, 2010). Given the lack of homology between these two elements, the most likely explanation is that both have co-opted a *rbc-1* sequence independently. When masking all sequence parts that are identified as blast *rbc-1* hits, and then creating and analysing the network, the cluster persists indicating that this is not the only sequence similarity within this TE group (Appendix Fig S4). We thus cannot exclude that the presence of the rbc-1 is a coincidence.

To investigate further the biological significance of this observation, we investigated the evolution of *rbc-1* and the Tc1/Fot1 TEs across the *Caenorhabditis* species. We found that *rbc-1* itself was invaded by Fot1 in common ancestor of the Elegans subgroup. *C. elegans* contains 3 Fot1 copies within *rbc-1* and none outside of this genomic region (Appendix Fig S5A). A similar configuration was found in most other species (Appendix Fig S5B); however, in *C. brenneri*, there has been a large expansion of this particular TE, resulting in many genomic copies outside of *rbc-1* (Appendix Fig S5B). A smaller expansion may also have taken place in *C. sinica* (Appendix Fig S5B).

As an additional example of how this network may be used to identify features of TEs, we wondered whether it could be used to detect potential horizontal transfer events where a TE from one species invades a second species (Gilbert & Feschotte, 2018). To test this, we simulated placing five copies of a *D. melanogaster* TcMar-Fot1 TE in the genome of *C. brenneri*. As expected, rerunning the network resulted in the formation of a new cluster containing sequences from the *D. melanogaster* TE subfamily with the transferred TE copies (Appendix Fig S6). Thus, horizontally transferred TEs could potentially be detected using our network approach.

Overall, these examples demonstrate the potential of our network approach to discover hitherto unappreciated properties of transposable element families. To further test the breadth of possible applications, we constructed a network using L1 retrotransposons in the human genome. Similar to the Tc1/mariner networks from invertebrate genomes, this formed several robust clusters (Appendix Fig S7A and B). Interestingly, a large proportion of these clusters contained more than one L1 subfamily (Appendix Fig S7C and D), potentially indicating the high degree of similarity between L1 subfamilies to recent activity (Beck *et al*, 2010). This illustrates the potential that our network approach could have broad applicability across metazoan genomes including mammals.

## Construction of a bipartite network to assess genome-wide TE content across metazoans

Transposable element content is highly variable across different genomes, and the reasons for this diversity are poorly understood. We developed a distinct approach to compare TE content across genomes using a bipartite network. This approach differs from the approach of Iranzo *et al* (2016) and our method to visualise Tc1/mariner TEs (Fig 1) because we are investigating TE content rather than the sequence similarity between TEs. In our network, nodes are either metazoan genomes across the phylogeny (Dataset EV1) or TE consensus sequences as identified by RepeatMasker and *de novo* identification of TEs using RepeatModeler (Dataset EV2). Only connections between TE nodes and genome nodes are allowed. The strength of the connection between TEs and genomes is a measure of the genome-wide abundance of the particular TE. We tested a variety of different measures of TE abundance: the count of TEs (Appendix Fig S8A), the total coverage in base pairs (Appendix Fig S8B), the coverage as a percentage of the genome (Appendix Fig S8C) and the coverage as a percentage of all TEs in the genome (Appendix Fig S8D). The four measures produced highly similar networks as measured by the rand index (Appendix Fig S8E). We therefore decided to focus on the most straightforward measure—the total coverage in base pairs, hitherto referred to as the "TE length" network. The network formed several well-defined clusters indicating a robust structure (Fig 3A and B). We assessed the contribution of different TE families to the network by systematically removing them and examining how key network statistics varied. This analysis identified Gypsy elements as particularly important in determining the network structure (Appendix Fig S9). We investigated this observation more closely to attempt to find potential explanations. Whilst Gypsy elements are found in all the genomes, most other TE families are also widely distributed so this is unlikely to be the reason why Gypsy is so important for network structure (Appendix Fig S9A). However, Gypsy has by far the largest number of nodes across the network, with over 10,000 connected sequences present compared with the next most abundant that has 2,500 (Appendix Fig S9B). This corresponds to a large total length, but there are other TEs with similar or higher total length (Appendix Fig S9C). Thus, the most likely explanation for the contribution of Gypsy to the network structure is the large number of nodes. Supporting this hypothesis, repeatedly removing 10% of the Gypsy elements selected at random resulted in perturbations to the network structure that were comparable to other TEs (Appendix Fig S9E–G, marked yellow).

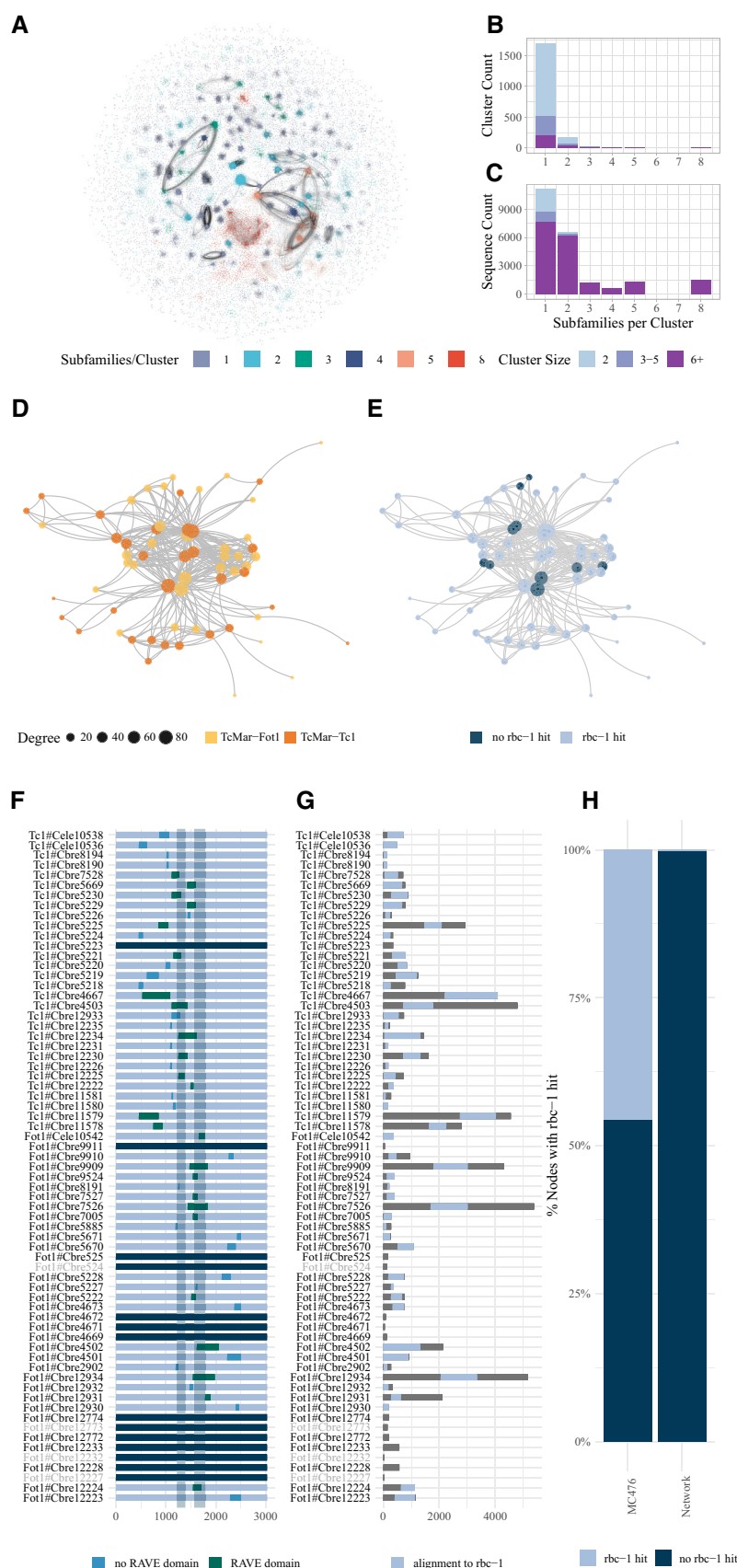

**Figure 2.**

**Figure 2. Further analysis of the sequence similarity network.**

A   Visualisation of the SSN using the openOrd layout algorithm with the nodes coloured by how many different subfamilies their cluster contains.
B   Histogram showing the number of clusters sorted by the number of subfamilies in each cluster. The shading indicates the number of sequences in each cluster.
C   Histogram showing the number of sequences that are in clusters with a certain number of subfamilies. The colours indicate cluster size.
D   Network visualisation of the mixed cluster mc476 using the Kamada–Kawai layout algorithm. The subfamily each TE sequence belongs to is highlighted by the node colour and the node size is proportional to the node degree.
E   Network visualisation of cluster mc476 using the Kamada–Kawai layout algorithm with nodes coloured according to the presence or absence of an *rbc-1* hit.
F   Diagram of all the Tc1 and Fot1 sequences and which part of the *rbc-1* sequence they matched with. Rbc-1 matching sequence parts are highlighted. Light blue indicates no match to the RAVE domain within the TE, and dark blue indicates a match to the RAVE domain within the TE. The dark grey vertical bars indicate the position of RAVE complex domains within the *C. elegans rbc-1* protein. The names of sequences not in the mixed cluster mc476 are printed in grey.
G   Diagram of all the Fot1 and Tc1 sequences with the alignment to *rbc-1* highlighted in light blue. The names of sequences not in the mixed cluster mc476 are printed in light grey.
H   Barplot showing the percentage of sequences with a *rbc-1* hit in the mc476 cluster and the whole network.

## piRNAs support higher diversity of TEs within organisms

We used our network to investigate how epigenetic silencing pathways might impact TE content across genomes. We annotated the genomes in our network according to whether they contained orthologues of DNMT1 and or DNMT3 (DNMTs), which are responsible for cytosine 5 DNA methylation in metazoan genomes (Law & Jacobsen, 2010) or the Argonaute Piwi proteins which are required for the presence of piRNAs (Weick *et al*, 2014). We used reciprocal blast searches and hmmer searches using the Pfam domains to annotate DNMTs and Piwi across the species in our network (Appendix Fig S10). Species containing DNMTs or piRNAs were evenly distributed across the network (Fig 3C and D). Species with DNA methylation tended to have more species-specific TEs (singletons—TE nodes that connect only to one genome node) than species lacking DNA methylation ($P < 0.001$) (Fig 4A). Species-specific TEs might be elements that evolved recently within one species. In contrast, they could be older TEs that are too divergent to be recognised as relatives of TEs in other species. Consistent with the latter possibility, the difference between species with DNMTs and those without was only seen for TEs lacking catalytic domains, which are likely to be older TEs that have lost activity (Appendix Fig S11A and B).

Species with piRNAs tended to a have higher degree than those species without piRNAs ($P < 0.001$; Fig 4B). The degree of the network measures the total number of TEs that are drawn from different families; thus, this is higher in species with piRNAs (Fig 4B). Interestingly, however this difference was not seen when we considered the weighted degree of genomes ($P = 0.013$), which is the sum of total number of counts of each TE (Fig 4C). This indicates that piRNA-containing species tended to have a greater diversity of TEs in their genomes but the average count of each TE was less.

A potential problem with this analysis is unequal sampling across taxonomic groups. For example, our network contains multiple fly genomes, all of which have Piwi but lack DNMT1 and DNMT3. Counting all these species may result in overrepresentation of the features of their genomes, which are due to their common ancestry rather than their TE silencing mechanisms. To circumvent this issue, we collapsed the species into nodes representing closely related species. As a result, each group of related species with the same silencing mechanisms (such as all flies) was counted only once. This produced a network with many fewer nodes (Fig 4E and F). Using this network, we investigated the connectivity of species groups with or without piRNAs or

DNMTs. The differences in singleton numbers between DNMT containing species and those without DNMTs were no longer significant ($P = 0.42$) (Fig 4G). However, species with piRNAs had a significantly higher degree compared with those without piRNAs ($P = 0.028$), indicating that this difference is still present once phylogenetic differences are controlled (Fig 4H and I). Thus, reduced diversity TEs evolved several times independently in species lacking piRNAs. Importantly, the total number of TEs and total coverage of TEs, linear measures of TE content that do not require network construction, only showed small differences between species with piRNAs and those without (Appendix Fig S12A and B). Furthermore, these differences disappeared after correcting for phylogeny (Appendix Fig S12C and D). Thus, examining network properties offers opportunities to detect effects of epigenetic silencing methods on TE diversity that would not be detectable through simple analyses of TE content.

Taken together, these analyses suggest that the control of TEs by piRNAs may allow increased diversity to accumulate in TEs within a genome, whilst the copy number of each TE remains low. This result is consistent with theoretical work investigating the effect of piRNA-mediated silencing on TE content as discussed further below (Lu & Clark, 2010).

## Discussion

Here, we describe a method to visualise TE content within genomes using networks. The generality of the network approach enabled us to use similar methods to analyse both the sequence content of TE families and the overall content of TEs from a variety of families across genomes. Both approaches enabled us to discover features of TEs across genomes.

First, by identifying co-clusters containing multiple subfamilies of TEs, we identified an interesting convergent evolution, whereby TEs from two Tc1/mariner subfamilies had co-opted sequences derived from *rbc-1* homologues. This prompted a more detailed examination of the evolution of Tc1/mariner elements associated with *rbc-1*, which showed that *rbc-1* was colonised by TEs specifically in the Elegans subgroup of *Caenorhabditis*. Within these organisms, the presence of TEs within *rbc-1* was highly conserved. *rbc-1* has a highly conserved housekeeping function; thus, we propose that, once TEs had invaded *rbc-1*, it was difficult for natural selection to remove them as deletions would be likely to damage the structure of the gene. This may explain the persistence of a small number of TEs

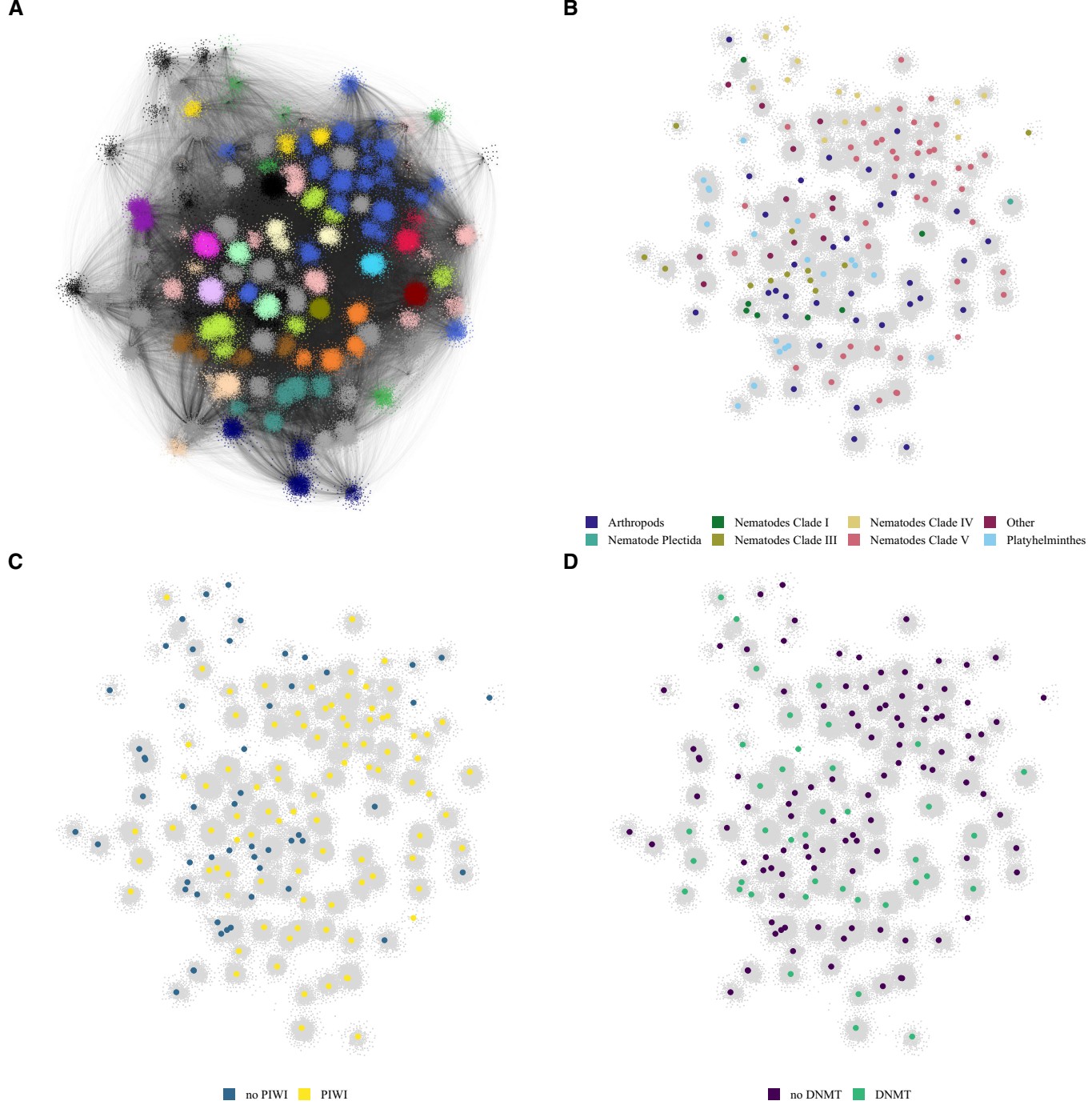

**Figure 3. Bipartite network visualisation.**

A  Visualisation of the bipartite network using the openOrd layout algorithm. The nodes are coloured according to their modularity class. Smaller clusters are coloured grey.

B  Visualisation of the bipartite network using the openOrd layout algorithm. All TE nodes are coloured in grey, and edges are omitted. The genome nodes highlighted by larger node size and coloured according to phylogeny.

C  Visualisation of the bipartite network using the openOrd layout algorithm. All TE nodes are coloured in grey, and edges are omitted. The genome nodes are highlighted by node size and coloured according to the presence and absence of PIWI genes.

D  Visualisation of the bipartite network using the openOrd layout algorithm. All TE nodes are coloured in grey, and edges are omitted. The genome nodes are highlighted by node size, species without DNMT1 or DNMT3 orthologues are purple, and species with an orthologue of either DNMT1 or DNMT3 are coloured green.

within *rbc-1*. These copies then act as a reservoir from which expansions may originate—this has occurred in *C. brenneri*, *C. dougherti* and, independently, *C. sinica*. In addition to the potential benefit to the TE from such a strategy, it is also possible that the colonisation of the *rbc-1* gene may have affected *rbc-1* gene function and comparative analysis between *rbc-1* in *Caenorhabditis* species where the colonisation did not occur would thus be of considerable interest.

Second, our bipartite network enabled us to establish differences in TE content that could not come from a one-dimensional analyses. We found that the presence of DNMTs is associated with a greater number of "Singleton" TEs that exist in only one genome. We showed that this is due to the presence of inactive TEs that are too divergent in sequence to be recognised as part of a wider family. Accumulation of inactive TEs in species with DNA methylation could be due to the increased mutation rate of methylated cytosine

compared with unmethylated cytosine (Bird, 1980). Alternatively, DNA methylation could suppress the negative consequences of inactive TEs, for example through restraining ectopic recombination between inactive elements, as has been shown to be a consequence of TE methylation in animals (Zamudio *et al*, 2015), thus enabling them to accumulate within genomes. Importantly, however the differences between species with DNA methylation and those without were no longer significant when we collapsed the network according to phylogeny. This may be due to lack of power due to a small number of independent losses of DNA methylation; thus, further work incorporating more species would be required to clarify whether this feature of DNA methylation-containing genomes is relevant generally or just in a few taxa.

We also discovered that species with the capacity to produce piRNAs had a higher degree, corresponding a greater diversity of

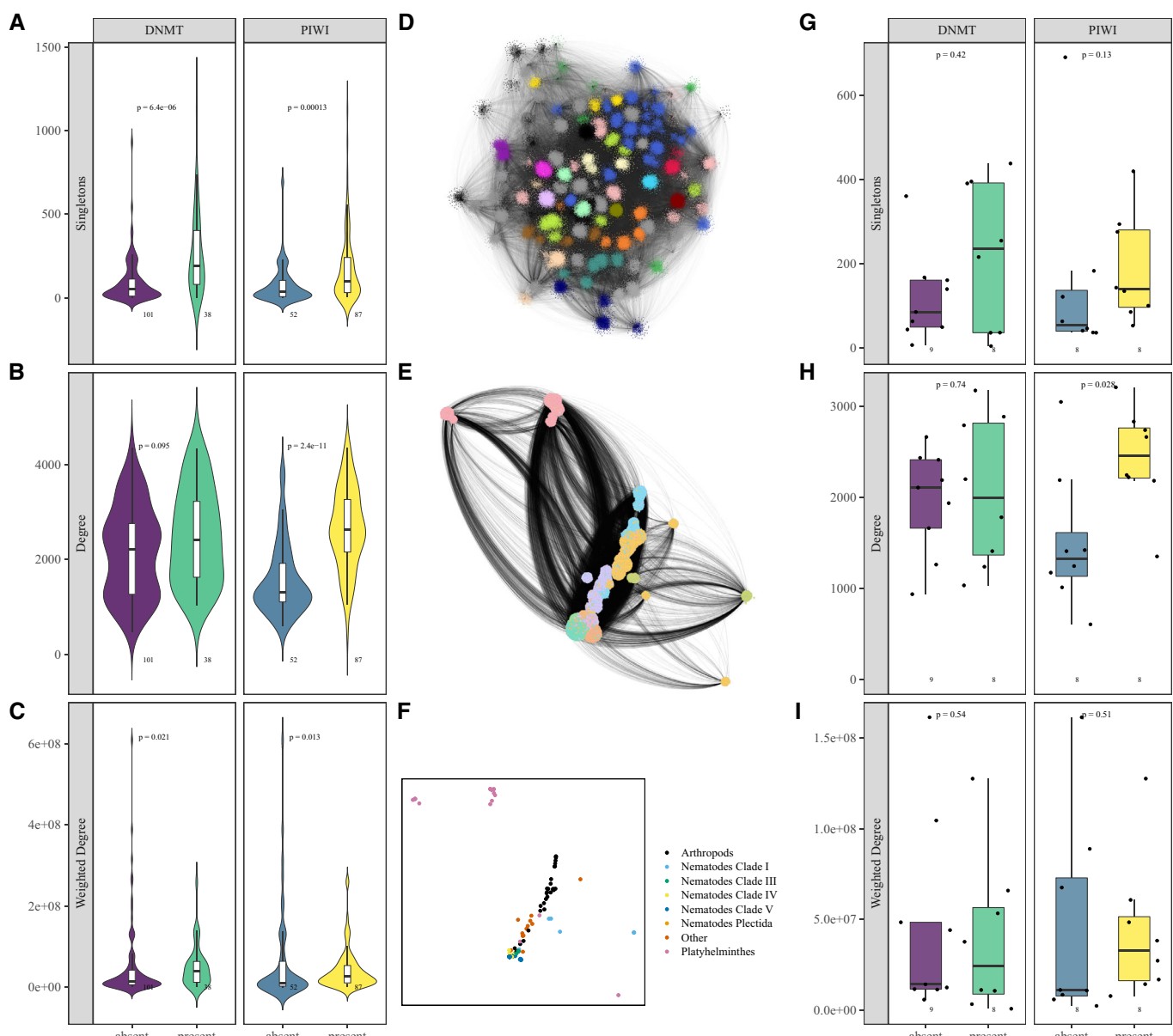

**Figure 4.**

**Figure 4.  Further analysis of the bipartite network.**

A–C   Violin plots and boxplots showing the bipartite network properties. In all figures, the box outlines the interquartile range (IQR) with the median shown as a line, whiskers indicate the fences (lower fence: lowest value at most Q1−1.5*IQR, upper fence: largest value no further than Q3+1.5*IQR) and data beyond the whiskers are drawn as individual points (outliers). There are 101 species with DNMTs, 38 species without DNMTs, 52 species with PIWIs and 87 species without PIWIs, as indicated below each box. (A) Number of unique TEs (singletons) in DNMT vs no DNMT species ($P < 0.001$, Wilcoxon) and PIWI vs no PIWI species ($P < 0.001$, Wilcoxon). (B) Degree (number of connections) in DNMT vs. no DNMT species ($P = 0.095$, Wilcoxon) and PIWI vs no PIWI species ($P < 0.001$, Wilcoxon). (C) Weighted degree (sum of the weights of the edges) in DNMT and no DNMT species ($P = 0.021$, Wilcoxon) and species with PIWI and without PIWI ($P = 0.013$, Wilcoxon).

D     Network visualisation of the bipartite network using the openOrd layout algorithm. Genome nodes are connected to TE reference nodes. The edge weight is proportional to TE coverage in base pairs. The nodes are coloured according to their modularity class. Smaller clusters are coloured grey. This is the same representation as Fig 3A, for ease of comparison.

E     Network visualisation of the orthonetwork using the openOrd layout algorithm. The nodes are genomes and TE reference sequences. Edges between genomes and TEs are proportional to TE content. Edges between genomes represent the number of shared orthogroups. The nodes are coloured according to their cluster.

F     Visualisation of all the clusters in the orthonetwork. Each node is one cluster coloured by the most prevalent phylogenetic classification of the species within the cluster.

G–I   Boxplots showing the orthonet network properties. In all boxplots, the box outlines the interquartile range (IQR) with the median shown as a line, whiskers indicate the fences (lower fence: lowest value at most Q1−1.5*IQR, upper fence: largest value no further than Q3+1.5*IQR) and data beyond the whiskers are drawn as individual points (outliers). There are 9 clusters without DNMTs, 8 clusters with DNMTs, 8 clusters with piRNAs and 8 clusters without piRNAs. (G) Number of singletons for DNMT ($P = 0.41$, Wilcoxon) and PIWI ($P = 0.13$, Wilcoxon). Each point is one cluster from the orthonetwork. (H) Node degree for DNMT ($P = 0.74$, Wilcoxon) and PIWI ($P = 0.031$, Wilcoxon). Each point is one cluster from the orthonetwork. (I) Weighted degree for DNMT ($P = 0.53$, Wilcoxon) and PIWI ($P = 0.49$, Wilcoxon). Each point is one cluster from the orthonetwork.

TEs. Interestingly, this did not correspond to an increased weighted degree, meaning that the total length of these TEs is not increased. Thus, species with piRNAs tend to have a larger number of TE families, each of which has fewer copies in the genome. One explanation for this is that the ability of each TE to spread is limited by the silencing activity provided by the piRNA pathway. This means that the copy number of the TE is directly restrained, and because the copy number remains low, it means that there is little selection pressure for natural selection to eliminate the TE. This has been suggested previously on the basis of population genetics simulations in one species (Lu & Clark, 2010), but our study is the first to show such a trend through studying TE content in extant species.

Overall, our use of both bipartite and monopartite networks with connection weights determined differently in each case demonstrates that network strategies offer considerable flexibility to create further networks to study TEs. For example, future applications could incorporate protein-coding genes to investigate protein-TE coevolution, and other genome metrics to gain new insights into how TEs shape genomes across evolution.

# Materials and Methods

## Identifying and classifying repeat sequences

We obtained 143 metazoan genomes from online databases (Dataset EV1). We then used RepeatMasker version 4.0.6 to identify and classify repeat sequences across these metazoan genomes (Smit *et al*, 2015). RepeatMasker uses the GIR Institute Repbase database of eukaryotic repeat sequences (Bao *et al*, 2015). To avoid biased detection and reporting of TE content due to unequal representation, we constructed a custom library of repeat elements following previously published methods (Szitenberg *et al*, 2016). We analysed all genomes with RepeatModeler for *de novo* repeat detection (Smit *et al*, 2015). Subsequently, we clustered all repeats into groups of 80% similarity by running the uclust algorithm to combine similar repeats and reduce redundancies (Edgar, 2010). To classify the repeats in our non-redundant library, we used

Censor (Kohany *et al*, 2006). Finally, we used this custom library of classified repeats (Dataset EV2) as the input library to run RepeatMasker on all genomes. After identifying repeat sequences, we filtered the data for TEs and excluded all unclassified TEs from further analysis.

## Sequence similarity network construction

To produce the sequence similarity networks, we identified all repeats that were classified by RepeatMasker as Tc1/mariner elements. We extracted their nucleotide sequence from the genomes using the boundaries provided by RepeatMasker. To establish sequence similarities between the repeats, we utilised the BLAST command line application blastn (Camacho *et al*, 2009). The BLAST algorithm is an approximation to the Smith–Waterman algorithm and used for local sequence alignment (Altschul *et al*, 1990). We then constructed the sequence similarity network with the Python package NetworkX 2.2 using the bitscore as the edge weight (Hagberg *et al*, 2008). For visualisation, we used Gephi (Bastian *et al*, 2009). To further analyse the mc476 cluster, we masked all rbc-1 blast hits in sequences within the cluster. Subsequently, we recalculated edge weights and redrew the network.

To construct the human LINE-1 sequence similarity network, we obtained the RepeatMasker table from the UCSC genome browser and extracted all sequences classified as LINE-1 from the human reference genome (GRCh38/hg38). Subsequently, we filtered the dataset to only include sequences with a minimum length of 500 base pairs and used blastn with an expect value threshold of $10e^{-30}$ to establish sequence similarities. We then constructed the sequence similarity network with the Python package NetworkX 2.2 using the bitscore as the edge weight (Hagberg *et al*, 2008). For visualisation, we used Gephi (Bastian *et al*, 2009).

## RAVE domain identification

All sequences from the mc476 cluster are from *Caenorhabditis* species (*C. elegans* and *C. brenneri*) so we performed a blastx search against the *C. elegans* proteome for each sequence in the

mc476 cluster. This identified rbc-1 as the most prevalent best hit (53 out of 63 sequences). We also used pfamscan to check all sequences for domains from the Pfam-A database (Punta *et al*, 2011). The most common match was the RAVE domain. rbc-1 has two RAVE complex protein Rav1 C-terminal domains. All 25 sequences with a RAVE domain have a best blastx hit to rbc-1.

## Bipartite network construction

We defined two node types, genome nodes and TE nodes. Each reference sequence from our custom library is a TE node, and each species is a genome node. Each TE found in a particular genome was connected to the relevant genome node with the edge weight corresponding to the TE length in basepairs. Several edge weight metrics were compared (Appendix Fig S8). We used this information to construct a bipartite network with NetworkX and visualised it using Gephi (Hagberg *et al*, 2008; Bastian *et al*, 2009). For the collapsed network, we analysed all genomes using orthofinder (Emms & Kelly, 2019). Subsequently, we created a new network with two node types: TEs and genomes. In this network, we have connections between species and TEs, which are weighted by an adjusted TE content metric. LengthAdj = log(TE length)/log(TE max_length)) species–species network. We also introduce edges between species pairs, to account for phylogeny. For the species–species edge weight, we determined the number of shared orthogroups between each species–species pair using orthofinder and introduced a threshold of 4,500 orthogroups to increase network sparsity and combat noise. The edge weight was calculated by dividing the number of shared orthogroups between two species by the maximum number of shared orthogroups in any pair. The edge weight adjustments are made to ensure both numbers are on the same scale.

## Network visualisation

To visualise the networks, we used Java with the Gephi Toolkit java library and the OpenOrd layout algorithm implemented in Gephi (Bastian *et al*, 2009; Martin *et al*, 2011). This algorithm was developed for large-scale undirected networks employing both force-directed layout and average-link clustering to better reflect the underlying data in the network layout. We used the same parameters for all created networks (Dataset EV3). We identified groups of sequences with high similarity by calculating the modularity statistic, using the Louvain Method for community detection (Blondel *et al*, 2008). This algorithm is implemented in Gephi as the modularity statistic and was run with edge weights, randomising and a resolution of one. Additionally, we identified connected components using the depth-first search algorithm (Tarjan, 1972). This algorithm is available in Gephi as the connected component statistic and was run for an undirected network. Both algorithms can be used to identify clusters within networks.

## Identification of the DDE3 transposase domain

We searched for the presence of a DDE3 transposase domain in the sequences present in the sequence similarity network. First, we translated all repeat sequences into amino acid sequences with EMBOSS transeq (Rice *et al*, 2000; Goujon *et al*, 2010).

Subsequently, we utilised the profile hidden Markov model method HMMER3 to find homologous sequences to the DDE3 family (PF13358) from the Pfam-A database via the hmmsearch utility (Eddy, 2011; Punta *et al*, 2011).

## Multiple sequence alignment

Multiple sequence alignment of the sequence similarity network cluster mc476 was performed using MUSCLE without any alignment curation. The phylogenetic tree was then constructed via maximum likelihood (PyML) and visualised with TreeDyn (Chevenet *et al*, 2006).

## Identification of potentially active singletons in the bipartite network

To investigate the relationship between singleton TEs and TE activity, we checked all singleton TEs for the following TE-associated Pfam domains via the hmmsearch utility (Eddy, 2011; Punta *et al*, 2011): DDE_1 (PF03184), DDE_2 (PF02914), DDE_3 (PF13358), retrotransposon gag protein (PF03732), integrase core domain (PF00665) and retroviral aspartyl protease (PF00077). If no domain was found, we called the singleton inactive, and if one or more of the domains were found within a TE sequences, we described the TE as active.

## Bootstrapping of singleton data

We sampled 30 genomes with DNMTs and calculated the p-value (Wilcoxon) between the number of active singletons and inactive singletons. This was repeated 1,000 times. The same was done for genomes without DNMTs. Subsequently, we compared the distribution of *P*-values.

## Ortholog prediction

We predicted orthologs for DNMT and PIWI proteins by identifying reciprocal best hits with BLAST using the human amino acid sequences [specify UniProt IDs] as the blastp query. For species with no reciprocal hit in the proteome, we used the tblastn BLAST tool to search the genome instead (Altschul *et al*, 1990). If there was a hit, we extracted it with additional 20 kb upstream and downstream sequences to account for introns. We then used exonerate to do pairwise sequence comparison with these sequences (Slater & Birney, 2005). We only considered predicted coding alignments with a minimum length of 100 aa, sequence similarity over 50% and a reciprocal best BLAST hit.

For all putative orthologs of DNMT and PIWI proteins, we identified in one of these two ways, we also used HMMR3 to investigate the presence of Pfam-A methylase and Pfam-A PIWI domains, respectively. To reduce the risk of false positives, we did BLAST searches of all putative orthologs against bacterial and plant proteomes (all SwissProt bacterial reference proteomes and the *Arabidopsis thaliana* proteome) and excluded putative orthologs that were due to contaminations. Only sequences with both a reciprocal best hit and a domain were considered for further analysis. This dataset was manually curated. We visualised our results using the EMBL online tool iTOL (Letunic & Bork, 2019).

**Distribution of rbc-1-associated TEs across *Caenorhabditis***

We used blastn to identify the genomic coordinates of blast hits (Stevens *et al*, 2019) to the members of the cluster mc476 from the sequence similarity network. We identified the location of homologues of *C. elegans* rbc-1 using Exonerate in –protein2genome mode with *C. elegans* rbc-1 and searched for mc476 homologues within rbc-1 using Bedtools.

## Data availability

All the code used in this manuscript is available via GitHub: https://github.com/lisa-london/TE-EpiEvo.

**Expanded View** for this article is available online.

### Acknowledgements
This work was funded by the Medical Research Council (Epigenetics and Evolution; to PS) and a MRC strategic doctoral studentship (to LS and PS).

### Author contributions
PS, DB and Y-KG conceived the study. PS, LS and DB designed the research. LS collected the data and constructed the networks with supervision by DB and PS. LS performed most analysis under supervision from PS with some supplementary analysis by PS. PS and LS drafted the manuscript; LS, PS, DB and Y-KG contributed to editing the manuscript.

### Conflict of interest
The authors declare they have no conflict of interest.

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
