## [Review Process File · Molecular Systems Biology]

Network-based visualisation reveals new insights into transposable element diversity

Lisa Schneider, Yi-Ke Guo, David Birch, and Peter Sarkies

DOI: [10.15252/msb.20209600](https://doi.org/10.15252/msb.20209600)

Corresponding author(s): Peter Sarkies (psarkies@imperial.ac.uk)

Review Timeline:

Submission Date:	5th Apr 20
Editorial Decision:	26th May 20
Revision Received:	13th Dec 20
Editorial Decision:	21st Dec 20
Revision Received:	26th May 21
Accepted:	27th May 21

Editor: Maria Polychronidou

Transaction Report:

Thank you again for submitting your work to Molecular Systems Biology. We have now heard back from the three referees who agreed to evaluate your study. Overall, the reviewers think that the presented findings seem potentially interesting. They raise however a series of concerns, which we would ask you to address in a major revision.

Without repeating all the points listed below, some of the more fundamental issues are the following:

- The methodology needs to be described in better detail and the differences (e.g. to the original approach by Koonin and colleagues) and advantages of the presented network approach over alternative strategies need to be clearly demonstrated.
- The biological findings need to be strengthened and expanded. Reviewer #3 provides constructive suggestions in this regard. Moreover, the significance of the presented findings needs to be better contextualized within existing knowledge.

All other issues raised by the reviewers should be convincingly addressed. Please let me know in case you would like to discuss any of the issues raised.

On a more editorial level, we would ask you to address the following.

REFEREE REPORTS

Reviewer #1:

In this manuscript, Schneider et al. use a network-based approach to analyse transposable element (TE) diversity in metazoan genomes. First, they build a monopartite sequence similarity network of Tc1/mariner sequences from five nematode genomes and identify a yet unknown connection between two transposon subfamilies, which suggests independent acquisition of a host protein domain. In addition, the authors construct a bipartite network from transposon sequences and genomes with edges proportional to TE abundance and suggest that genomes that have piRNA machinery feature a wider variety but lower copy number of TEs.

In principle, networked-based visualization can be very useful for understanding TE evolution and

diversity between genomes. However, the rationale for using such networks in the present study is not clear, the presented analyses and their results are inadequately described, and the conclusions are poorly discussed.

Specific comments:

- 1) The authors claim that they have "developed a method to track TE evolution using network analysis". However, it is not clear what are the new developments compared to the previously reported method from Iranzo, Krupovic and Koonin. As described, the study rather seems to be an application of the above method to a different subject.
- 2) The authors built a monopartite network based on pairwise sequence comparison of transposon sequences. What is the advantage of using such a network over canonical phylogenetic tree reconstruction? In the case of the Tc1/mariner family, the standard phylogenetic approach would have probably yielded the same results. It seems to me that using domains as nodes within a bipartite network could have been a better strategy to represent the complex evolutionary history of transposons.
- 3) What is the authors motivation for studying Tc1/mariner elements in nematodes. This TE group has been extensively studied in the past. Some claims on missing information in the field of TE evolution and diversity (in Introduction) also seem overstated in light of the available literature (see eg. page 3, lines 21-24).
- 4) It is unclear which genomes and TE classes did the authors use for the bipartite network reconstruction. The "SupplementalTable" file contains an unformatted list of names, which may be the list of genomes used, but it is not referenced anywhere in the main text or figures. Also, Supplemental Figure 6 is confusing. What do the colored symbols indicate? If one of them marks the presence of PWI, that would indicate that PWI-containing genomes cluster together. If this is the case, how can the effect of the piRNA pathway on transposon diversity be entangled from the contribution of other genetic factors? Although the authors seem to acknowledge these problems and try to tackle them by 'collapsing' closely related genomes to a single node, I am not sure how this would remove potential correlation between phylogeny and the presence of the piRNA machinery.
Furthermore, there seems to be a problem with PWI annotation. For example, according to the Supplemental Figure 6 (if I understand it correctly), no PWI protein has been found in *C. elegans*, which is one of the model organisms for studying the piRNA pathway. Due to these confusions it is difficult to judge the significance of the correlation between the presence of a piRNA pathway and transposon abundance.
At the same time, I also wonder what would be the advantage of reconstructing such a complicated network to link TE content and piRNA machinery? Simple correlation analysis would do the job.
- 5) A thoughtful discussion of the findings is generally missing in the manuscript and the implications of specific results are poorly explained. For example, what does the "higher connectivity of Tc1 elements with a DDE3 domain" (page 5 line 4-5) mean and what does this imply? What is the *rbc-1* gene and why could independent acquisition of *rbc-1* be important? What are RAVE domains and what do they do? Also, how is the effect of Gypsy elements on network structure relevant and why do genomes with DNMT have more species-specific TEs?

Minor points:

- Specific aspects of the study design are poorly justified and explained. Why did the authors focus on analysing the Tc1/Fot1 cluster? Why did they choose the TE length network, etc.?
- Figure legends are generally poor and Fig 3A in particular would deserve a more useful representation.
- Figure panels are not cited in order.

Reviewer #2:

Schneider and colleagues present a network-based method to visualize diversity and content of transposable elements (TEs) across several genomes.

TEs play central roles in genome evolution. Understanding how regulatory mechanisms impact TE diversity and the trends of TE accumulation or decay are key questions in modern evolutionary genomics. While sequencing technologies evolved immensely to provide hard data for such studies, the methods to test these questions across broader phylogenetic scales or within large datasets are still scant.

Schneider and colleagues put forward a valuable contribution to these type of studies. They show that network-based approaches can be very useful to identify correlational trends between the evolution of regulatory mechanisms and the qualitative and quantitative content of TEs in genomes. This opens an exciting venue to investigate these trends, and how and if they impact the evolutionary patterns observed in these lineages.

While I am not in a technical position to be able to judge the algorithms used for network reconstruction, I believe the work to have great value and I am strongly supportive of its publication.

There are however, a few things about the manuscript that can be improved. In particular, using the text to define each concept under analysis will help broaden the readership of their work.

Minor points

Usage of the terms appears to be inconsistent, sometimes leading to confusion. Examples: 'connectivity' versus 'degree of contributing to the network': is it the same thing? And if yes, it is perhaps best to be consistent on how to call it across the manuscript.

Page 5, lines 2-5: perhaps use the term autonomous and non-autonomous Tc1 copies.

Figure 3A: what are the colors representing?

Some figures labels could be more descriptive, helping readers that are not so familiar with the analysis of networks to understand what is shown. e.g. Figure 4D: 'singletons' or 'degree' (of what?), 'weighted degree'. What are the units each Y axis is representing?

What does it mean to 'contribute to a larger degree to the network'? Defining this on text may facilitate interpretation of the data by non-specialists.

Place the p-values after each mention on the text might help link described analysis with observed effects. E.g. page 7, lines 10-12: 'Whilst the differences between DNMT containing species and those without DNMTs were no longer significant ($p=0.41$), there was a significantly increased number of TEs in species with piRNAs compared to those without piRNAs ($p=0.13$).'

Main points

Page 5, lines 15-25: It is puzzling what would be the explanation for the clustering of Fot1 with Tc1

subfamilies found in the same node by the monopartite network analysis. While the monopartite network approach allowed to identify it, further inspection showed that the putative convergent co-option of a *rbc-1* cannot explain the clustering (as clustering was maintained upon re-analysis using the dataset with the *rbc-1*-derived sequences masked). Without any further explanation or discussion it's hard to conclude that the network analysis was really able to point towards the discovery of these novel properties of these subfamilies.

There are also several (hundreds) clusters encompassing 2 or more Mariner/Tc1 subfamilies (Figure 2a). What made the authors focus on this particular one? The text/material and methods provides insufficient information to understand if these were logical steps or if the finding of *rbc-1*-derived RAVE domains in this particular cluster was serendipitous.

Could the authors please elaborate on this? This is important for readers to understand how they can use this network-based approach to attempt to identify novel properties of TE families as it was done in this section of the manuscript.

Page 6, lines 10-11: What exactly is the 'TE length network' and why focusing on this? It does not match any of the tested 'measures of TE abundance' mentioned just before in the text (those being: TE count, % coverage of all genome, % coverage of all TEs, and total coverage in bp). This might be stated in a confusing way if the terms for the same parameter change constantly. Please check consistency in case they are the same.

Page 6, lines 12-15: Which metazoan genomes from which taxonomical categories were included in the bipartite network? Are they all included in the phylogeny shown in Supplemental Figure 6? It is not described anywhere else in the manuscript except in the legend on Figure 3d (albeit at the phylum level; 3 phyla + "other"). Detailing this is important because the interesting finding that Gypsy elements contribute the most to the network structure might reflect taxonomical biases of this analysis towards Ecdysozoans. This should be brought to attention in the text/discussed and perhaps highlighted graphically in Supplemental Figure 6.

Page 7, lines 4-5: Why taxonomical bias and resulting "pseudoreplication" is a concern when trying to identify trends in connectivity among species with different combination of TE-defense systems, but not before?

Page 8, lines 1-2 in the discussion: perhaps it would be interesting to elaborate more into why multipartite networks offer an advantage over one-dimensional analysis in the case presented here. Specially since comparative genomics and ANOVA/correlation testing of distribution of characters onto a phylogeny are standard approaches to attempt to reveal some of these trends.

I apologize for the delays in peer review due to unforeseen problems on our side, and congratulate the authors on their manuscript.

With my best regards,

Joan Barau
IMB - Mainz

Reviewer #3:

Summary and General Remarks:

In this manuscript Schneider et al. follow a novel approach to look at properties of transposable elements in several species. The authors have been inspired by a similar network approach used to analyze the evolution of viruses. This network-based analysis reveals previously uncharacterized and unexpected findings about transposable elements. Although, the title of the manuscript refers to the evolutionary analysis of metazoans this analysis is carried out across 5 invertebrate genomes.

The reason for selecting Mariner/Tc1 superfamily for the initial illustration of the method has not been described. Nevertheless, this monopartite network study reveals an independent co-option of an *rbc-1* sequence in a cluster of 2 subfamilies (Fot1 and Tc1). This unexpected finding demonstrates a potential benefit of the network analysis and clustering, but the impact or importance of such a finding is left without discussing further in the text and therefore seems only descriptive.

The authors then move on to test their approach using a bipartite network to assess genome-wide transposable element content. This network is further used to address the role of epigenetic silencing pathways in transposable element content in genomes. First, the authors briefly describe the impact of DNA methylation. Then they look into the presence of piRNAs. These epigenetic silencing pathways appear to have contributed to a larger variation of repeats but also to less copy numbers of individual transposable elements.

The two halves of the manuscript have not been conceptually connected, which makes it a difficult read. In order to present a coherent study, an additional transition section should be added to the text connecting the two halves.

There could be a general interest to this study by the evolutionary biologists especially those working on understanding the contribution of transposable elements to genome evolution. However, the authors need to work on increasing the breadth of this study.

Major points:

If possible, a network analysis in some of the vertebrate genomes could be important to include to demonstrate the robustness of the used approach.

The following questions should be addressed:

1. What could be an evolutionary advantage of the acquisition of the RAVE domain? Is there a functional link or evidence? What is the significance of this discovery?
2. Why are the Gypsy elements important in determining the network structure? The Figure S5 would definitely benefit from having a better representation and description. In the current state it is very difficult to understand what this figure means.
3. The analysis reveals that the species which have DNA methylation end up with more of species-specific transposable elements. What is the possible explanation for that? Is the silencing by DNA methylation a driving force for diversification of transposable elements?
4. Could this approach or a similar approach be used for investigating the co-evolution of protein families (e.g. KRAB-ZFPs) and transposable element subfamilies? If yes, it would be an important outlook to discuss in the end of the paper. Similarly, could this approach detect horizontally transferred transposable elements?

Minor points:

1. Page numbers are missing.
2. There is no Figure title for Figure 2
3. The colors used for Arthropods and Nematodes Clade V in Figure 3B are too close to distinguish from each other in a printed-out version. Same is true for Figure 4C.
4. Figure 4D is referred to in the text before Figure 4A, B, C and a re-organization of figure panels might help solve this issue.
5. In general, the figures could be tweaked to make it more appealing what is being presented. For example, the data labels are not aligned in Figure 3 and there is an overlap of p-values with data points in Figure 4.
6. On page 8 line 7, there's a full stop in the middle of the sentence.

Overall, this paper could be suitable for publication upon major revision.

Reviewer #1:

In this manuscript, Schneider et al. use a network-based approach to analyse transposable element (TE) diversity in metazoan genomes. First, they build a monopartite sequence similarity network of Tc1/mariner sequences from five nematode genomes and identify a yet unknown connection between two transposon subfamilies, which suggests independent acquisition of a host protein domain. In addition, the authors construct a bipartite network from transposon sequences and genomes with edges proportional to TE abundance and suggest that genomes that have piRNA machinery feature a wider variety but lower copy number of TEs.

In principle, networked-based visualization can be very useful for understanding TE evolution and diversity between genomes. However, the rationale for using such networks in the present study is not clear, the presented analyses and their results are inadequately described, and the conclusions are poorly discussed.

Specific comments:

1) The authors claim that they have "developed a method to track TE evolution using network analysis". However, it is not clear what are the new developments compared to the previously reported method from Iranzo, Krupovic and Koonin. As described, the study rather seems to be an application of the above method to a different subject.

>>The methods that we developed to investigate TE evolution was indeed inspired by the approach taken by Iranzo et al., but both the sequence similarity method and the TE content network are very different in the way they were constructed from the Iranzo et al., method. The Iranzo et al., method constructed a bipartite network between viral proteins and in the sequence alignment network (Figure 1,2) the network is monopartite where each node is a different Mariner family TE and the strength connecting two nodes is the sequence identity between the two. The TE content network differs again from the Iranzo method because here no sequence information is used as the strength connecting the species nodes to the TE nodes is the abundance of that particular TE within the genome of that species. The networks that we have constructed are therefore so different in form and content from the Iranzo approach that it is not appropriate to describe it as an application of the same method to a different subject.

In order to emphasize this point clearly in the manuscript we have added 2 paragraphs explicitly comparing our methods to the Iranzo method, one referring to the sequence alignment network and one to the TE content network. done

2) The authors built a monopartite network based on pairwise sequence comparison of transposon sequences. What is the advantage of using such a network over canonical phylogenetic tree reconstruction? In the case of the Tc1/mariner family, the standard phylogenetic approach would have probably yielded the same results. It seems to me that using domains as nodes within a bipartite network could have been a better strategy to represent the complex evolutionary history of transposons.

>>Our approach offers two clear advantages over a traditional phylogenetic tree reconstruction. The first is that even within the Tc1/mariner family the diversity is such that phylogenetic trees are poorly resolved with very low branch support values. For an example of this we refer the reviewer to the tree that we constructed using only a small part of the network- the TcMar/Fot1 elements from the cluster analysed in Figure 2, which is shown in Figure EV3 of the revised submission. Even within these relatively closely related TEs the branch support values are low, meaning that the confidence in inferring anything about the evolutionary history of these TEs is low.

More generally our approach offers a different set of information, that can be complementary to a more linear approach based on phylogenetic reconstruction.

We have added a section in the manuscript to explicitly point out how phylogenetic reconstruction would be of limited use in this case. *done*

3) What is the authors motivation for studying Tc1/mariner elements in nematodes. This TE group has been extensively studied in the past. Some claims on missing information in the field of TE evolution and diversity (in Introduction) also seem overstated in light of the available literature (see eg. page 3, lines 21-24).

>>We chose this family to illustrate our method because

- i) As the reviewer points out, it has been well studied in *C. elegans*, thus at least in this species it was a good test case for our ability to recover and classify Tc1/mariner elements.**
- ii) The fact that Tc1 elements are well characterised enabled us to search for the presence of catalytic residues within them, which would not have been possible in a less well understood element.**
- iii) Our study included genomes from very distant species of nematodes within which Tc1/mariner elements are not so well studied, offering the opportunity for comparison across a wide range of evolutionary distances.**

We believe that the value of our approach was illustrated by the novel connection between Rave elements and a subset of Tc1/mariner TEs, which had not been observed previously despite the extensive characterisation of these TEs in nematodes.

4) It is unclear which genomes and TE classes did the authors use for the bipartite network reconstruction. The "SupplementalTable" file contains an unformatted list of names, which may be the list of genomes used, but it is not referenced anywhere in the main text or figures.

>>We apologise for the lack of clarity here. The supplemental table is the list of genomes and we have now improved this table so it includes information about the source of the assembly used and referenced this in the manuscript.

Also, Supplemental Figure 6 is confusing. What do the colored symbols indicate? If one of them marks the presence of PIWI, that would indicate that PIWI-containing genomes cluster together. If this is the case, how can the effect of the piRNA pathway on transposon diversity be entangled from the contribution of other genetic factors? Although the authors seem to acknowledge these problems and try to tackle them by 'collapsing' closely related genomes to

a single node, I am not sure how this would remove potential correlation between phylogeny and the presence of the piRNA machinery.

>>The purpose of “collapsing” in this case is to ensure that multiple closely related species with the same Piwi/DNMT status are counted only once, thus correcting for phylogenetic effects. We have now included an expanded explanation of this in the main text.

Furthermore, there seems to be a problem with PIWI annotation. For example, according to the Supplemental Figure 6 (if I understand it correctly), no PIWI protein has been found in *C. elegans*, which is one of the model organisms for studying the piRNA pathway. Due to these confusions it is difficult to judge the significance of the correlation between the presence of a piRNA pathway and transposon abundance.

>>We are not certain how the reviewer came to this conclusion- there is a clear symbol for the presence of Piwi in *C. elegans* in this figure (now Figure EV9)

At the same time, I also wonder what would be the advantage of reconstructing such a complicated network to link TE content and piRNA machinery? Simple correlation analysis would do the job.

>>The network properties showed a significant difference but more straightforward properties such as TE content show very little difference, showing that the network properties reveal information that is masked by simpler methods of annotation. This is now shown as a supplemental figure (Figure EV11).

5) A thoughtful discussion of the findings is generally missing in the manuscript and the implications of specific results are poorly explained. For example, what does the "higher connectivity of Tc1 elements with a DDE3 domain" (page 5 line 4-5) mean and what does this imply?

>>We were reluctant to overinterpret any of our findings but we do agree that we could provide some pointers as to possibilities and so we have added an extended interpretation to this section. The higher connectivity of DDE3-domain containing Tc1s means that they have a larger number of closely related TEs within the network than Tc1s with no match to the DDE3 domain. The presence of the DDE3 domain is required for TE activity and therefore TEs with intact DDE3 domains are more likely to be active. A possible interpretation of the higher connectivity is therefore that TEs with the DDE3 domain are more recently active and therefore will have been descended from a TE at a different site more recently.

What is the *rbc-1* gene and why could independent acquisition of *rbc-1* be important? What are RAVE domains and what do they do.

>>The *C. elegans* *rbc-1* domain is homologous to yeast Rav1 and human dmx1. The RAVE domain is a key, conserved region in the c terminus of this protein, which is involved in the formation of the RAVE complex which, in both human and yeast, assembles the V-type ATPase. Though the function of the *C. elegans* homologue has not been explicitly studied, given the function of the yeast and human homologues it is likely that it is also involved in assembly of the V-type ATPase. It is therefore a highly conserved housekeeping gene. We speculate that the insertion of a TE within the protein is a “safe hiding place” for the TE because natural selection is unlikely to remove the TE as the function of *rbc-1* is likely too important. Supporting this, in the revised version of the manuscript we provide an analysis of the conservation of the association between the *rbc-1* protein and the Tc1/Fot1 elements (Figure EV5), showing that the association emerged specifically in the *C. elegans* subgroup of Caenorhabditis and has been maintained throughout; interestingly there have been expansions of the element in *C. brenneri* where copies are found in the genome outside of the *rbc-1* protein.

Also, how is the effect of Gypsy elements on network structure relevant

>>We have added an exploration of why the contribution of gypsy elements is so important. Removing gypsy elements has a very large effect on network structure and we now show that this is because the gypsy elements are the most abundant across the entire network- if, instead of removing all gypsy elements, we remove 10% of them, then it no longer has such a large effect. We have added a supplemental figure to this effect (Figure EV8).

Why do genomes with DNMT have more species-specific TEs?

>>We do not have a definitive answer to this question, but in the revised version we tested whether the effect of DNMT presence or absence on our network was driven by active (presumed younger) or inactive (presumed older) TEs. The effect was significantly greater when only inactive TEs were considered (now included as supplemental figure). We raise two possible hypotheses to explain this. First, as it is known that DNA methylation is mutagenic, it is possible that TEs in genomes with DNA methylation evolve faster due to targeting of DNA methylation, and therefore increased mutation rate, such that they appear to be species-specific due to the absence of close relatives. Second, it is possible that DNA methylation counteracts potentially harmful effects of inactive TEs (such as ectopic recombination) meaning that they persist in the genome for longer evolutionary time, thus accumulating mutations that make them more distant from the ancestral TE and thus appear species-specific.

Minor points:

- Specific aspects of the study design are poorly justified and explained. Why did the authors focus on analysing the Tc1/Fot1 cluster?

>>This was because it was the largest cluster with at least one TE subfamily. This has been clarified in the manuscript.

Why did they choose the TE length network, etc.?

>>TE length is straightforward to assess and easy to interpret. This has been clarified in the manuscript done

- Figure legends are generally poor and Fig 3A in particular would deserve a more useful representation.

We have revised the figure legends to further clarify them.

- Figure panels are not cited in order.

We have corrected this.

Reviewer #2:

Schneider and colleagues present a network-based method to visualize diversity and content of transposable elements (TEs) across several genomes.

TEs play central roles in genome evolution. Understanding how regulatory mechanisms impact TE diversity and the trends of TE accumulation or decay are key questions in modern evolutionary genomics. While sequencing technologies evolved immensely to provide hard data for such studies, the methods to test these questions across broader phylogenetic scales or within large datasets are still scant.

Schneider and colleagues put forward a valuable contribution to these type of studies. They show that network-based approaches can be very useful to identify correlational trends between the evolution of regulatory mechanisms and the qualitative and quantitative content of TEs in genomes. This opens an exciting venue to investigate these trends, and how and if they impact the evolutionary patterns observed in these lineages.

While I am not in a technical position to be able to judge the algorithms used for network reconstruction, I believe the work to have great value and I am strongly supportive of its publication.

There are however, a few things about the manuscript that can be improved. In particular, using the text to define each concept under analysis will help broaden the readership of their work.

Minor points

Usage of the terms appears to be inconsistent, sometimes leading to confusion. Examples: 'connectivity' versus 'degree of contributing to the network': is it the same thing? And if yes, it is perhaps best to be consistent on how to call it across the manuscript.

>> Thank you for this point. The two are the same thing. We have ensured that “connectivity” is used consistently.

Page 5, lines 2-5: perhaps use the term autonomous and non-autonomous Tc1 copies.

Figure 3A: what are the colors representing?

>> the nodes are coloured by the modularity cluster they belong to. Smaller clusters are coloured grey. We expanded the figure legends to explain this.

Some figures labels could be more descriptive, helping readers that are not so familiar with the analysis of networks to understand what is shown. e.g. Figure 4D: 'singletons' or 'degree' (of what?), 'weighted degree'. What are the units each Y axis is representing?

>> singletons are unique TEs only found in one genome. We included an explanation in the text and figure legend.

What does it mean to 'contribute to a larger degree to the network'? Defining this on text may facilitate interpretation of the data by non-specialists.

>>We have clarified this in the text

Place the p-values after each mention on the text might help link described analysis with observed effects. E.g. page 7, lines 10-12: 'Whilst the differences between DNMT containing species and those without DNMTs were no longer significant ($p=0.41$), there was a significantly increased number of TEs in species with piRNAs compared to those without piRNAs ($p=0.13$).'

>> We added p-values into the paper.

Main points

Page 5, lines 15-25: It is puzzling what would be the explanation for the clustering of Fot1 with Tc1 subfamilies found in the same node by the monopartite network analysis. While the monopartite network approach allowed to identify it, further inspection showed that the putative convergent co-option of a *rbc-1* cannot explain the clustering (as clustering was maintained upon re-analysis using the dataset with the *rbc-1*-derived sequences masked).

Without any further explanation or discussion It's hard to conclude that the network analysis was really able to point towards the discovery of these novel properties of these subfamilies.

There are also several (hundreds) clusters encompassing 2 or more Mariner/Tc1 subfamilies (Figure 2a). What made the authors focus on this particular one? The text/material and methods provides insufficient information to understand if these were logical steps or if the finding of *rbc-1*-derived RAVE domains in this particular cluster was serendipitous.

Could the authors please elaborate on this? This is important for readers to understand how they can use this network-based approach to attempt to identify novel properties of TE families as it was done in this section of the manuscript.

>>We thank the reviewer for these points. We focussed on the Tc1/Fot1 cluster because it was the largest cluster containing two or more subfamilies. We have now clarified this in the text. We sought to explain why the two subfamilies cluster together and found that many TEs within the cluster had alignments to *rbc-1*. It is interesting that the cluster still persists after the *rbc-1* alignment has been removed, which may indicate that there is convergent sequence similarity in other regions of the protein; thus to some extent the link between the cluster and the *rbc-1* domain is serendipitous. Nevertheless, we would not have identified the similarity between these two subfamilies on the basis of traditional phylogenetic analyses because they are not closely related.

In order to explore the biology behind this connection in more detail, we now include an analysis of the genomic location of these elements across all Caenorhabditis. We find that the colonisation of the *rbc-1* protein in the common ancestor of the elegans subgroup is the reason for the acquisition of the RAVE domain within the Fot-1 TE. In most species there are a small number of TEs, all located within the *rbc-1* protein. In *C. brenneri*, there has been a large expansion of this TE family from its location within the *rbc-1* protein.

Page 6, lines 10-11: What exactly is the 'TE length network' and why focusing on this? It does not match any of the tested 'measures of TE abundance' mentioned just before in the text (those being: TE count, % coverage of all genome, % coverage of all TEs, and total coverage in bp). This might be stated in a confusing way if the terms for the same parameter change constantly. Please check consistency in case they are the same.

>> We apologise for this lack of clarity. The TE length represents the total coverage in basepairs and we have clarified this in the manuscript.

Page 6, lines 12-15: Which metazoan genomes from which taxonomical categories were included in the bipartite network? Are they all included in the phylogeny shown in Supplemental Figure 6? It is not described anywhere else in the manuscript except in the legend on Figure 3d (albeit at the phylum level; 3 phyla + "other"). Detailing this is important because the interesting finding that Gypsy elements contribute the most to the network structure might reflect taxonomical biases of this analysis towards Ecdysozoans. This should be brought to attention in the text/discussed and perhaps highlighted graphically in Supplemental Figure 6.

>> We have added explanation about the choice of genomes. We have also added analysis explaining why the Gypsy element contributes most to the network structure. Essentially the abundance of the Gypsy element across all species means that removing it has a very large effect on the network structure. When we remove only 10% of gypsy elements, more in line with the abundances of other elements across the network, the effect on network structure is smaller, thus supporting our interpretation. (Figure EV8)

Page 7, lines 4-5: Why taxonomical bias and resulting "pseudoreplication" is a concern when trying to identify trends in connectivity among species with different combination of TE-defense systems, but not before?

>> We used a two-step process to examine this- first we wanted to use the complete network as a platform to discover potentially significant relationships, because there is much more information giving more power to detect relationships. Subsequently we wanted to test whether these relationships might be independent of phylogeny. We believe this two-step process maximises the chances of discovering and validating important relationships which might be missed if focussing only on the minimal collapsed network.

Page 8, lines 1-2 in the discussion: perhaps it would be interesting to elaborate more into why multipartite networks offer an advantage over one-dimensional analysis in the case presented here. Specially since comparative genomics and ANOVA/correlation testing of distribution of characters onto a phylogeny are standard approaches to attempt to reveal some of these trends.

>> We thank the reviewer for this suggestion. We believe that our experimental method offers a complementary approach to traditional phylogeny as it provides a much more intuitive visualisation, and also does not suffer from the difficulty in reconstructing phylogenetic trees of transposable elements. We are not arguing that our approach should replace traditional phylogenetic approaches. We now include these points in the discussion of the manuscript.

I apologize for the delays in peer review due to unforeseen problems on our side, and congratulate the authors on their manuscript.

With my best regards,

Joan Barau
IMB - Mainz

Reviewer #3:

Summary and General Remarks:

In this manuscript Schneider et al. follow a novel approach to look at properties of transposable elements in several species. The authors have been inspired by a similar network approach used to analyze the evolution of viruses. This network-based analysis reveals previously uncharacterized and unexpected findings about transposable elements. Although, the title of the manuscript refers to the evolutionary analysis of metazoans this analysis is carried out across 5 invertebrate genomes.

The reason for selecting Mariner/Tc1 superfamily for the initial illustration of the method has not been described. Nevertheless, this monpartite network study reveals an independent co-option of an *rbc-1* sequence in a cluster of 2 subfamilies (Fot1 and Tc1). This unexpected finding demonstrates a potential benefit of the network analysis and clustering, but the impact or importance of such a finding is left without discussing further in the text and therefore seems only descriptive.

The authors then move on to test their approach using a bipartite network to assess genome-

wide transposable element content. This network is further used to address the role of epigenetic silencing pathways in transposable element content in genomes. First, the authors briefly describe the impact of DNA methylation. Then they look into the presence of piRNAs. These epigenetic silencing pathways appear to have contributed to a larger variation of repeats but also to less copy numbers of individual transposable elements.

The two halves of the manuscript have not been conceptually connected, which makes it a difficult read. In order to present a coherent study, an additional transition section should be added to the text connecting the two halves.

There could be a general interest to this study by the evolutionary biologists especially those working on understanding the contribution of transposable elements to genome evolution. However, the authors need to work on increasing the breadth of this study.

Major points:

If possible, a network analysis in some of the vertebrate genomes could be important to include to demonstrate the robustness of the used approach.

>> Thank you for the suggestion- we decided instead to take the approach of selecting species from arthropods and doing the same analysis on Tc1/mariner because this would allow us to use the same custom libraries for our analysis. This is now included as a supplemental figure (Figure EV 2). The trends were mostly recapitulated in this network, demonstrating that it can be used more broadly outside nematodes.

The following questions should be addressed:

1. What could be an evolutionary advantage of the acquisition of the RAVE domain? Is there a functional link or evidence? What is the significance of this discovery?

>> In order to provide further information we have characterised the association between the rbc-1 protein and the Tc1 and Fot1 families that cluster together across the Caenorhabditis species. We find that the ancestral state was likely colonisation of Tc1 and Fot1 within the rbc-1 protein, as most species retain this configuration. We speculate that due to the essential function of the rbc-1 protein it represents a “safe harbour” for TEs as TEs within this protein are unlikely to be removed by deletion of the gene. In *C. brenneri*, and to a lesser extent in *C. sinica* and *C. dougherti*, the TEs have expanded out of the rbc-1 protein to colonise the genome. A figure illustrating the prevalence of the members of this cluster across the species is now included as Figure EV5 and we have expanded the results section with a discussion of this point.

2. Why are the Gypsy elements important in determining the network structure? The Figure S5 would definitely benefit from having a better representation and description. In the current state it is very difficult to understand what this figure means.

>> We believe that the Gypsy elements are crucial for the network structure because they are the most widespread throughout the network. In support of this, the perturbation of the network that occurs when all gypsy elements are removed is much greater than when only 10% are removed. We now include this as an additional Figure (Figure EV8)

3. The analysis reveals that the species which have DNA methylation end up with more of species-specific transposable elements. What is the possible explanation for that? Is the silencing by DNA methylation a driving force for diversification of transposable elements?

>> We have performed further analysis to test the hypothesis that DNA methylation might be associated with diversification of TEs. We subdivided species specific TEs into active and inactive elements on the basis of the presence of TE domains required for

activity (see materials and methods). Our analysis revealed that inactive, species-specific TEs show an increased prevalence within species with DNA methylation whereas those containing domains are not. This suggests that DNA methylation is associated with accumulation of inactive TEs. (Figure EV10)

4. Could this approach or a similar approach be used for investigating the co-evolution of protein families (e.g. KRAB-ZFPs) and transposable element subfamilies? If yes, it would be an important outlook to discuss in the end of the paper. Similarly, could this approach detect horizontally transferred transposable elements?

>> We thank the reviewer for these interesting suggestions.

We have used the monopartite network to investigate the possibility that our approach could detect horizontally transferred transposable elements, by spiking in TEs from Drosophila into the C. brenneri genome. This showed that the artificially inserted TEs formed a cluster containing Drosophila TEs, demonstrating that horizontally transferred TEs could in principle be detected on the basis of their anomalous position within the network. This has been added as a new supplemental figure (Figure EV6). We agree with the reviewer that it would be possible to investigate protein-TE coevolution using our network approach and we have included a comment on this in the manuscript.

Minor points:

1. Page numbers are missing.

2. There is no Figure title for Figure 2

>> done

3. The colors used for Arthropods and Nematodes Clade V in Figure 3B are too close to distinguish from each other in a printed-out version. Same is true for Figure 4C.

>> changed to colourblind friendly and printer friendly colourscale

4. Figure 4D is referred to in the text before Figure 4A, B, C and a re-organization of figure panels might help solve this issue.

>> order changed

5. In general, the figures could be tweaked to make it more appealing what is being presented. For example, the data labels are not aligned in Figure 3 and there is an overlap of p-values with data points in Figure 4.

>> fixed

6. On page 8 line 7, there's a full stop in the middle of the sentence.

>>Corrected

Overall, this paper could be suitable for publication upon major revision.

Thank you for sending us your revised manuscript. We have now heard back from the three reviewers who were asked to evaluate your study. Overall, the reviewers think that the study has improved as a result of the performed revisions. However, reviewer #1 still raises several remaining concerns. As you probably already know, our editorial policy is to in principle allow a single round of major revision. Nevertheless, given the supportive comments provided by reviewers #2 and #3, in combination with the fact that the remaining concerns of reviewer #1 seem addressable, we would like to offer you a chance to address these remaining issues, as well as some more minor ones raised by reviewer #3, in an exceptional (and last) round of revision. It is very important that the remaining issues are convincingly addressed.

Without repeating all the points listed below, the most substantial concern raised by reviewer #1 is the recommendation to perform analyses in vertebrate genomes. Given the moderate novelty of the study with respect to the methodology, we agree with reviewer #1 that such analyses would enhance the confidence in the broader relevance of the approach. As such, we would strongly encourage you to perform them. All other issues raised would need to be convincingly addressed.

On a more editorial level, we would like to ask you to address the following issues.

REFEREE REPORTS

Reviewer #1:

In the revision, the authors substantially improved the clarity of the manuscript. Methodological descriptions have been improved and some new analyses and discussions have been added. It is clear that the use of networks (particularly the bipartite network) has merit for analysing TE distribution and evolution and the work presents some interesting new findings. However, some previous comments of the referees remained unaddressed and the revisions pose new questions.

1. With regard to the novelty of the methodology, I do understand that the methods constitute new elements and features compared to the original strategy used by Koonin and colleagues. However, I still think that it is more appropriate to describe them as an advancement of an existing method, rather than a conceptually novel method. Thus, I advise that the authors refrain from using terms like "we develop a novel method".
2. The added analyses extend the breadth of the study, but some work in vertebrate genomes would have been highly valuable to demonstrate utility of the approach beyond invertebrates.
3. I am now somewhat confused about the role of the *rbc-1* domains in the clustering of Tc1/Fot1 elements in the monopartite network. First, the authors suggest that clustering is due to independent *rbc-1* gene acquisition, but later they show that masking out *rbc-1* does not affect the network topology. This makes me wonder if the monopartite networks can be considered generally useful for identification of convergently evolved domains or the identification of *rbc-1* is rather an accidental finding?
4. To illustrate the advantage of the monopartite network over conventional phylogenetic reconstruction, the authors built a phylogenetic tree with members of Tc1 and Fot1 subfamilies. Why were other elements not included in this reconstruction?
5. On a related note, Fot1 elements are typically found in fungi, and I could not find reports of Fot1 TEs in nematodes. If this is the first discovery of a Fot1 element in these organisms, that should be clearly stated and confirmed. Or could it be that the subgroup named Fot1 here is in fact a subclade of Tc1 that was misannotated?
6. As requested earlier, it would be helpful if the authors explain their choice of studying Tc1/mariner elements within the manuscript.
7. Similarly, with regard to independent acquisition of RAVE domains, please comment on the potential benefit of this domain for the TEs.
8. The title of the manuscript should be revised: it refers to an analysis "across metazoans", but the work was carried out in selected invertebrate genomes.
9. I must also point out that the response letter was rather poorly structured and I struggled to find the mentioned edits in the manuscript text. The addition of respective page/line numbers for each

revision and/or tracking the changes in different colour would have been very helpful.

Reviewer #2:

I believe the authors have responded adequately to the points raised during revision, substantially improving the manuscript. As a result I believe the manuscript is suitable for publication.

Reviewer #3:

Schneider et al. has definitely improved the manuscript upon revision. More specifically; the clarifications on the novel aspects of their analysis and the extended interpretations of the findings on *rbc-1* and on the contributions of epigenetic modifiers to TE diversities have made this manuscript a much more interesting read. Moreover, the quality of the figures has substantially increased. The new additions into the "Discussion" section makes the reader curious to discover the great potential of this type of network analysis.

In this current form the manuscript appears to be suitable for publication.

Nevertheless, I'd like to suggest a few changes in the figures in order to improve visualization:

1. It has escaped my attention in the first submission that in Figure 2 panel b, the light color used for cluster size larger than 5 is invisible in the printed version. It could be simply solved by adding line-borders to the bars.
2. In Figure 4, panel f: some of the nodes are far away and outside of the panel territory. Therefore, a box around the clusters could help the reader not miss those far away nodes.

1
2
3
4
5
6
7
8
9
10
11
12
13
14
15
16
17
18
19
20
21
22
23
24
25
26
27
28
29
30
31
32
33
34
35
36
37
38
39
40
41
42
43
44
45

➤ *Reviewer 1 comments*

1. With regard to the novelty of the methodology, I do understand that the methods constitute new elements and features compared to the original strategy used by Koonin and colleagues. However, I still think that it is more appropriate to describe them as an advancement of an existing method, rather than a conceptually novel method. Thus, I advise that the authors refrain from using terms like "we develop a novel method".

>we have removed the reference to “novel” as requested.

2. The added analyses extend the breadth of the study, but some work in vertebrate genomes would have been highly valuable to demonstrate utility of the approach beyond invertebrates.

>We thank the reviewer for this point and now include an analysis of the L1 element in the human genome using our method. We describe this in the manuscript as follows:

“To further test the breadth of possible applications we constructed a network using L1 retrotransposons in the human genome. Similar to the Tc1/mariner networks from invertebrate genomes this formed several robust clusters (Appendix Figure S7A,B). Interestingly a large proportion of these clusters contained more than one L1 subfamily (Appendix Figure S7C,D), potentially indicating the high degree of similarity between L1 subfamilies to recent activity(Beck *et al*, 2010). This illustrates the potential that our network approach could have broad applicability across metazoan genomes including mammals. “

46
47
48
49
50
51
52
53
54
55
56
57
58
59
60
61
62
63
64
65
66
67
68
69
70
71
72
73
74
75
76
77
78
79
80
81
82
83
84
85
86
87
88
89
90
91
92
93

3. I am now somewhat confused about the role of the *rbc-1* domains in the clustering of Tc1/Fot1 elements in the monopartite network. First, the authors suggest that clustering is due to independent *rbc-1* gene acquisition, but later they show that masking out *rbc-1* does not affect the network topology. This makes me wonder if the monopartite networks can be considered generally useful for identification of convergently evolved domains or the identification of *rbc-1* is rather an accidental finding?

➤ This is possible and we have indicated this in the manuscript.

“We thus cannot exclude that the presence of the *rbc-1* is a coincidence.”

4. On a related note, Fot1 elements are typically found in fungi, and I could not find reports of Fot1 TEs in nematodes. If this is the first discovery of a Fot1 element in these organisms, that should be clearly stated and confirmed. Or could it be that the subgroup named Fot1 here is in fact a subclade of Tc1 that was misannotated?

➤ We thank the reviewer for highlighting this point. This was the annotation that came from our pipeline- we would not go beyond this in stating categorically that this is a novel discovery of Fot1s in nematodes but we now highlight that it has not been documented previously as follows:

“As an example, we selected one of these clusters, mc476, which was composed of two subfamilies annotated by our pipeline as Fot1 and Tc1 (Figure 2D). Fot1 TEs are most common in fungi, although some metazoan examples have been found (Dupeyron *et al*, 2020). The origin of this element in nematodes is thus unknown.”

5. As requested earlier, it would be helpful if the authors explain their choice of studying Tc1/mariner elements within the manuscript.

➤ We thank the reviewer for this point and have now included a sentence to this effect in the manuscript as follows:

“We chose this TE family due to its wide phylogenetic distribution (Dupeyron *et al*, 2020), and because Tc1/mariner elements are well characterised as active TEs in *C. elegans* (Bessereau, 2006).”

6. Similarly, with regard to independent acquisition of RAVE domains, please comment on the potential benefit of this domain for the TEs.

➤ This was in the discussion of the revised version, and we highlight it again here for the avoidance of doubt:

“*rbc-1* has a highly conserved housekeeping function thus we propose that, once TEs had invaded *rbc-1*, it was difficult for natural selection to remove them as deletions would be likely to damage the structure of the gene. This may explain the persistence of a small number of TEs within *rbc-1*. These copies then act as a reservoir from which expansions may originate- this has occurred in *C. brenneri*, *C. dougherti* and, independently *C. sinica*. In addition to the potential benefit to the TE from such a strategy, it is also possible that the colonisation of the *rbc-1* gene may have affected *rbc-1* gene function and comparative analysis between *rbc-1* in *Caenorhabditis* species where the colonisation did not occur would thus be of considerable interest.”

94
95
96
97
98
99
100
101
102
103
104
105
106
107
108
109
110
111
112
113
114
115
116
117
118
119
120
121
122
123
124
125
126
127
128
129
130
131
132
133
134
135
136
137
138

7. I must also point out that the response letter was rather poorly structured and I struggled to find the mentioned edits in the manuscript text. The addition of respective page/line numbers for each revision and/or tracking the changes in different colour would have been very helpful.

>We apologise, **line numbers have been added to this rebuttal.**

Figures & Figure legends:

1. After the request of our data analyst, you write in Figure EV7A: "This is the same representation as in Figure 3A, for ease of comparison." We think that this has been erroneously pasted there instead of the legend for Figure EV7B, can you please correct?
This has now been corrected
2. It has escaped my attention in the first submission that in Figure 2 panel b, the light color used for cluster size larger than 5 is invisible in the printed version. It could be simply solved by adding line-borders to the bars.
We changed the colours so this is no longer a problem
3. In Figure 4, panel f: some of the nodes are far away and outside of the panel territory. Therefore, a box around the clusters could help the reader not miss those far away nodes.
This is now included in Figure 4
4. Please provide the description of each EV Datasets in a separate tab in the corresponding .xls files. The Datasets descriptions should only be provided in the datasets themselves and need to be removed from the manuscript text.
This is now done
5. Dataset EV2 is provided as a PDF file, it should be provided as .xls file instead.
Now xls
6. Dataset EV3 has been provided twice, please remove the duplicate and make sure only the correct one is uploaded.
Corrected.
7. Due to the large number of EV Figures, we would ask you to include all of them in a PDF called Appendix. Appendix Figures should be labeled and called out as: "Appendix Figure S1, Appendix Figure S2, ... etc.". Each Appendix Figure legend should be provided below the corresponding Figure in the Appendix. Please include a Table of Contents in the beginning of the Appendix. For detailed instructions regarding expanded view please refer to our Author Guidelines: <http://msb.embopress.org/authorguide#expandedview>.
We now include an appendix as requested

Thank you again for sending us your revised manuscript. We are now satisfied with the modifications made and I am pleased to inform you that your paper has been accepted for publication.

Corresponding Author Name: SARKIES

Manuscript Number: MSB-20-9600